# Challenges and Strategies of Successful Mentoring: The Perspective of LEADS Scholars and Mentors from Minority Serving Institutions

**DOI:** 10.3390/ijerph18116155

**Published:** 2021-06-07

**Authors:** Patricia Y. Talbert, George Perry, Luisel Ricks-Santi, Lourdes E. Soto de Laurido, Magda Shaheen, Todd Seto, Deepak Kumar, Alexander Quarshie, Maya Thakar, Doris M. Rubio

**Affiliations:** 1College of Nursing and Allied Health Sciences, Howard University, Washington, DC 20059, USA; 2Department of Neurobiology, The University of Texas at San Antonio, San Antonio, TX 78249, USA; George.Perry@utsa.edu; 3Cancer Research Center, Hampton University, Hampton, VA 23669, USA; luisel.rickssanti@hamptonu.edu; 4Research Institute for Global Health Promotion and Health Education, School of Health Professions, University of Puerto Rico–Medical Sciences Campus, San Juan 00921, Puerto Rico; lourdes.soto1@upr.edu; 5Department of Internal Medicine, College of Medicine, Charles R. Drew University of Medicine and Science, Los Angeles, CA 90059, USA; magdashaheen@cdrewu.edu; 6The Queen’s Medical Center, University of Hawaii John A. Burns School of Medicine, Honolulu, HI 96813, USA; tseto@queens.org; 7Julius Chambers Biomedical Biotechnology Research Institute (BBRI), Department of Pharmaceutical Sciences, North Carolina Central University, Durham, NC 27707, USA; dkumar@nccu.edu; 8Department of Community Health and Preventive Medicine, Morehouse School of Medicine, Atlanta, GA 30310, USA; aquarshie@msm.edu; 9Institute for Clinical Research Education, University of Pittsburgh School of Medicine, Pittsburgh, PA 15213, USA; MST41@pitt.edu

**Keywords:** Leading Emerging and Diverse Scientists to Success (LEADS), faculty mentoring, faculty development, minority serving institutions (MSIs)

## Abstract

Mentoring continues to be a salient conversation in academia among junior and senior faculty and administrators. Mentors provide guidance and structure to junior faculty so that they can meet their academic and professional goals. Mentors also convey skills in balancing life and academic pursuits. Therefore, the purpose of this descriptive study was to provide additional insight from a training program called Leading Emerging and Diverse Scientists to Success (LEADS) regarding successful strategies and challenges of mentoring relating to lessons learned from the scholars and mentees’ perspective. The LEADS program provided multiple training platforms to increase skills and knowledge regarding research to promote expertise in grant writing and submission for funding opportunities among diverse scientists. These findings reinforce the knowledge about the value of a mentor in helping define the research pathway of their mentee and underscoring the importance of mentoring.

## 1. Introduction

While career development and mentoring programs are not new concepts in higher education, innovative approaches that increase the success of the next generation of underrepresented (UR) academicians are needed. African Americans/Blacks, Hispanics, Native Americans, Alaska Natives, Native Hawaiians, and Native Pacific Islanders make up almost 30 percent of the US population, but only account for 9% of Science, Technology, Engineering, and Mathematics (STEM) faculty at US colleges and universities [1]. Further, even when people who are UR enter academia, too many do not advance along the professorial pathway at the same rate as individuals from non-UR racial and ethnic groups, and many eventually leave academia [2,3].

It is well known that strong networks of support within and outside the institution [4], formal mentoring by successful UR faculty [5], and networking with others who can provide advice [6] enable the advancement of junior faculty and postdoctoral fellows. Yet, amid countless success stories, UR individuals frequently experience challenges in their mentoring relationships for a variety of reasons. Lack of time is a frequent challenge mentors face. For example, at minority serving institutions (MSIs), faculty frequently have heavy teaching loads and face the challenge of not having enough time for research, professional development, as well as mentoring. In addition, department chairs and administration at MSIs that are less-resourced institutions often are reluctant to provide release time for mentoring. Junior UR faculty without mentors are often challenged with not having adequate publications for a grant and not having time to work on those valuable publications. In addition, we know that investigators of color are significantly less likely than their White counterparts to receive NIH funding [7,8].

To this end, the Leading Emerging and Diverse Scientists to success (LEADS) program was established at the University of Pittsburgh in 2015 through a grant from the National Institutes of Health (R25 GM116740). We forged a partnership with nine MSIs (i.e., Charles Drew University, Hampton University, Howard University, Meharry Medical College, Morehouse School of Medicine, North Carolina Central University, John A. Burns School of Medicine, University of Puerto Rico, and the University of Texas at San Antonio), working with a senior leader at each site and focusing on junior faculty and postdoctoral fellows wanting to launch their research careers. LEADS focused on developing the skill set of early career investigators while simultaneously providing mentoring and coaching from trained senior faculty, as well as resources that allowed them to partake in research dissemination, networking opportunities, and additional training. In another paper, we describe LEADS in detail [9].

Since the establishment of the program, the LEADS administration collected data that explore the benefits and barriers related to mentoring. Through an iterative process, the program was modified as we applied lessons learned over the years. In this manuscript, we describe lessons learned, successful strategies, and provide insight from the participants’ perspective. We also explore the impact of the COVID-19 pandemic on mentoring. We are using descriptive research methods as required by the nature of the data we have.

### 1.1. Challenges of Mentoring, Especially in MSIs

Collectively, the authors have noticed that mentoring programs at MSI face numerous challenges, such as limited time from senior faculty, finding mentors whose area of research aligns with the mentee, and competing demands on mentees’ time to establish their research. COVID-19 compounded the problem as research labs were closed and access to patients was impossible. These challenges are even more relevant for mentoring programs focused on UR faculty at MSIs. For instance, UR junior faculty at MSIs have reduced access to funded investigators for mentoring as there are not as many at well-resourced institutions. They also have increased teaching demands with heavier course loads. 

### 1.2. LEADS’ Approach to Mentoring

The LEADS program has been described in detail elsewhere [9]. One particular aspect deserves additional focus, however: LEADS’ coaching and mentoring components. Incorporating coaching into mentoring is key to successful mentoring. Mentor and coaching training put both the mentee and mentor on solid footing, promoting shared goals, allowing mentors to function as sponsors, and supporting the careers of mentees. The LEADS program provided 2-day career coaching training to the site leaders, with the expectation that they would serve as career coaches and mentors for the scholars throughout the program. The site leaders found considerable value in this training and asked for additional training, which we offered the subsequent year. Coaching consisted of establishing relationships between site leaders and LEADS scholars. The site leaders also attended group training sessions and engaged with grant writing experts who provided a comprehensive overview of every stage of the grant writing process so that they could provide purposeful mentoring to their respective LEADS scholars. 

The LEADS program provided a forum to share best practices across institutions, particularly around mentoring. The program also augments the success of MSIs senior leaders and mentors who are often burdened with leadership, research, and instruction responsibilities. Equally important is the networking that occurs among scholars in the purposefully planned LEADS environment, resulting in cross-institutional collaboration. Collaboration among MSIs promotes institutional excellence to enrich their respective communities. LEADS does just this by not trying to transform the MSIs into a micro-University of Pittsburgh, but rather to have each institution live up to its vision and embrace their strengths. 

## 2. Methods

The purpose of this descriptive study was to provide additional insight, from the LEADS scholars’ perspective, regarding the challenges to and strategies of successful mentoring learned through the LEADS program. In short, the LEADS program was a one-year fellowship where scholars participated in online modules focused on developing their research. We also had an annual Summit for all of the LEADS scholars and site leaders for more intensive training and networking. In addition, scholars were expected to meet with the site leader at their MSI who provided mentoring and career coaching. Site leaders were also responsible for recruiting scholars for the program. 

### 2.1. Participants

Each year, the LEADS program solicits applications from the participating MSIs. Applications were also advertised across other MSIs, particularly those institutions that had a Research Center in Minority Institutions (RCMI). Over the five years, the range of applications we received was 14–27. Once we received the applications, the Institute for Clinical Research Education (ICRE) Diversity Committee at the University of Pittsburgh reviewed the applications for appropriateness for the program. Applicants were reviewed based on their readiness to embark on a research career and time commitment to fully engage in LEADS. The program required that participants have 20% protected time for the program. Each applicant was asked to upload a letter from their department chair documenting their protected time.

Each year we accepted 13–17 scholars. We tried to keep enrollment limited to 15 as that was the maximum number that the instructors could accommodate without diluting the impact of the training. Participant characteristics are provided under the results.

### 2.2. Study Procedures

To understand the impact of this training program, the LEADS Administration Team surveyed the scholars at baseline (the start of the program) and then annually thereafter. Each scholar received an email with a unique link to the online survey. They were sent reminder emails every week for five weeks. 

### 2.3. Measures

We created a survey to collect data on scholars’ characteristics related to research skills, research experience, mentoring structure, number of times meeting with the mentor, methods of communication with the mentor, and effectiveness of the mentor. Other data were collected on other factors such as burnout, motivation, and leadership. However, for the purposes of this study, we are only focusing on mentoring.

For this descriptive study, we have self-reported data from the LEADS scholars. In addition to the site leaders, many of the scholars had mentors in their area of research. Scholars responded to the mentoring questions with a focus on their primary mentor or mentoring team. 

## 3. Results

### 3.1. Population Characteristics

We analyzed baseline data for 61 scholars from five cohorts that participated in the LEADS program over five years and the exit survey data from 26 scholars (Table 1). The median age of the participants was 36 years old (IQR = 33–44 years) and the majority (78%) were female. About 36% were of Hispanic or Latino ethnicity. For race, 50% were African American, 17% were Caucasian, and 10% were Asian. More than one-third of the participants (34%) were tenure track, and 3.3% were tenured; scholars had a median annual percent effort in research of 50% (IQR = 30–90%), had a median of 8 years (IQR = 5–12 years) in research, and more than two-thirds of them (67%) had a goal of being independent research investigators (Table 1). In informal conversations with the scholars at one of the Summits, scholars indicated that they experienced barriers to successful mentoring including: unstructured mentorship relationships; lack of available experienced mentors in the same area of interest as them; inability to meet regularly; inaccessible mentors; lack of transparency in communication and poor communication; passive approach to mentoring; and no clear expectations of mentees. We did not collect qualitative data.

### 3.2. Mentoring

Table 2 represents the descriptive analysis of the scholars’ self-reported responses related to mentoring structure and effectiveness of mentorship at the baseline survey compared to the exit survey. The common structure of mentorship was a mentoring team with a primary mentor (baseline = 37.7% and exit survey = 50%). The majority of the scholars reported meeting with their mentors 0–3 times per month (baseline = 75.4% and exit = 84.6%).

Overall, most of the scholars (70%) rated their mentor as effective/very effective at baseline, which increased to 76% at the exit survey. Scholars rated their mentors’ advice about balance between professional and personal life as effective/very effective (baseline = 47% and exit = 64%) and career development and balancing professional responsibility (baseline = 61% and exit = 85%), and receiving help in developing research skills (baseline = 60% and exit = 77%). The scholars agreed or strongly agreed that their mentors were accessible (baseline = 83% and exit = 84%), provided useful critique of their work (baseline = 90% and exit = 95%), and motivated them to improve their work product (baseline = 80% and exit = 81%). For the mentoring team, scholars agreed or strongly agreed that the team contributed and effectively communicated more than one mentor to their professional development (baseline = 64% and exit = 77% and baseline = 64% and exit = 69%, respectively) and provided research knowledge and skills (baseline = 69% and exit = 77%). 

## 4. Discussion

Through the engagement in online modules and annual Summit, scholars learned how to become effective academic researchers. They participated in grant writing workshops, team science training, and cohort development activities to foster networking. In addition, the site leaders met with their respective scholars to provide career coaching and mentoring. Additionally, each scholar had their own mentor in their research content area.

Unfortunately, LEADS scholars and mentors at MSIs were often burdened with heavy teaching loads or disproportionate administrative duties, resulting in inadequate time for research, professional development, and mentoring. Demands placed on junior faculty often preclude them from spending significant time on research endeavors, hampering efforts in remaining relevant in their field. COVID-19 has disrupted the mentoring relationship [10]. As a result, the mentoring experience has been difficult and is likely causing anxiety regarding delayed career advancement. 

Despite these challenges, the majority of LEADS scholars rated the effectiveness of their mentor as high, which improved at the exit survey. In fact, many of the questions regarding mentoring improved at the exit survey. One reason that might explain this is the focus of the online modules in helping them drive their own careers and navigate the mentoring relationship. Though the LEADS program, scholars were able to connect with the teaching faculty of the modules, LEADS leadership, and site leaders. This expanded their mentoring network and could explain the more positive responses at the exit survey. 

The frequency of meeting with the mentor also increased at the exit survey as compared to baseline, as did the accessibility of the mentors. One explanation could be that the scholars learned how to make better use of their mentors through the LEADS program. The online modules are focused on the different components needed to launch a successful career. Therefore, as scholars become more knowledgeable, they become empowered to interact on a different level with their mentors. One area where this is seen is in their rating of the mentors’ effectiveness in helping them develop research skills. Only 19% rated them as very effective at baseline, but 46% rated them very effective at the exit survey. This dramatic increase could be a result of the scholars being able to engage with the mentors in a more focused way with the new knowledge they gained through the program. 

There are countless articles on mentoring and how it impacts mentees careers. It is common knowledge that good mentoring can promote the careers of mentees. In a study on mentoring indigenous mentees, Murry et al. found that mentorship differs across cultures [11]. This is an important finding in that this descriptive study is focused on the mentoring culture within MSIs. We found that empowering the mentees at MSIs to drive their own careers can improve the mentoring relationship. 

These findings can be used by other academic institutions, particularly MSIs, if they are looking to improve their mentoring programs. This study suggests that by training postdoctoral fellows and junior faculty and providing career coaching, these scholars learn how to drive their own careers and more effectively work with their mentors. 

### 4.1. Barriers to Mentoring during the Coronavirus (COVID-19) Pandemic

On 30 January 2020, the World Health Organization declared the coronavirus (COVID-19) a global health emergency and a global pandemic on 11 March 2020 [12]. The pandemic has impacted nearly every aspect of our academic and scientific lives and has posed significant challenges to the early stage investigator. It has become increasingly clear that the impact of COVID-19 on mentoring will likely be profound and long-standing. 

The most common barrier to mentoring is also possibly the most underappreciated. It is difficult to be a mentor or a mentee during an existential crisis. Faced with a generational health and financial crisis, it not surprising that nearly 50% of the general population report anxiety or depressive symptoms, with 32% reporting concern about finances and 80% being at least slightly worried about their employment [13]. The psychological impact of COVID-19 on students has also been reported. Son and colleagues interviewed 249 college students in Texas one month following a stay-at-home order and found that over 70% reported increased stress and anxiety, with 89% reporting difficulty concentrating and 86% reporting disturbed sleep patterns [14]. Similarly, in a study of 530 medical students who were quarantined for 2 weeks, 44% showed emotional detachment from friends and family, 24% felt depressed, and 38% were emotionally drained [15].

Another set of barriers are related to the technological advances that were rapidly adopted as alternatives to face-to-face meetings. Virtual meetings, virtual private networks (VPNs), voice over Internet protocols (VoIPs) and cloud technology have made remote work and distance learning feasible, even as it increases the risk of social isolation, lower work performance, demotivation, disengagement, and conflicts between work and family as we work from home [16,17]. Early stage investigators and junior faculty are particularly at risk for experiencing anxiety about delays in career advancement due to reduced visibility and opportunities for networking [17,18] Although access to digital resources and broadband internet is a health disparities issue that has substantially worsened during the COVID-19 pandemic, the impact of this “digital divide” on early stage investigators, particularly from minority serving institutions, has not been well-described. 

The third set of COVID-19-related barriers to mentoring are more system-related. COVID-19 has forced many universities and health care organizations to re-prioritize and institute work guidelines for health and safety. For some investigators, long-term experiments have been halted or terminated, with reduced or frozen start-up funding and suspended hiring of research staff to meet budgetary restrictions. Scientists and laboratory staff may be restricted from performing their work on-site or re-deployed to assist in other high-need areas [19]. Overall, these barriers may have precluded access to laboratories, routine library visits, or opportunities to join together face-to-face and engage socially with each other, but it will not stop the process of exploring and being a part of the research community. Researchers, scientists, mentees, and many other scholars are adopting new practices to allow them to continue exploring their research interests. 

### 4.2. Limitations

This study has several limitations. First, it did not use any theoretical framework to inform the descriptive study. Since this is a descriptive study, we are not positing causal relationships, which a theoretical framework could inform. Second, LEADS is a career development program for postdoctoral fellows and junior faculty at MSIs. People opt to apply to the program. The program is somewhat competitive, so we do not know if this would help all postdoctoral fellows and junior faculty at MSIs. However, we have had five cohorts of scholars over five years and the results are consistent. Finally, we did not survey the mentors, so we do not know what their perspective is of mentoring or how their mentoring has changed since COVID. However, many of the authors of this manuscript are mentors and have provided their perspective. 

## 5. Conclusions

The LEADS program partnered with nine MSIs to train their postdoctoral fellows and junior faculty so that they could launch a successful research career. The LEADS scholars engaged in online modules that were focused on topics such as team science, grant writing, and critical and creative thinking. In addition, the scholars participated in an annual summit where special speakers gave a variety of presentations all focused on advancing their research careers. In addition, the LEADS scholars received career coaching from licensed career coaches who worked one on one with the scholars to plan their careers. Site leaders and mentors dedicated time to working with the LEADS scholars to guide them through their careers. Through this descriptive study, we have shown that mentoring can improve as junior faculty are trained to better navigate the mentoring relationship and have learned how to receive better mentoring from their mentors. This indicates that professional development is a critical component that positively impacts mentoring, particularly at MSIs where scholars have limited access to mentors. With COVID-19, many additional challenges surfaced as faculty moved to sheltering at home. This limited access to mentors, as scholars would not see their mentors in the hall nor could they walk to their office. Anxiety and depression also significantly increased, which added additional challenges for scholars to navigate. However, these results suggest that a program, such as LEADS, can improve mentoring by training the scholars how to navigate the mentoring relationship better. Additionally, with career coaching, they learned how to drive their own careers. 

## Figures and Tables

**Table 1 ijerph-18-06155-t001:** Population characteristics of the scholars from the baseline survey (n = 61).

Characteristic	n ^a^	% ^b^
Age (median, 25th–75th percentile)	36	33–44
Female	46	78%
Race		
American Indian or Alaskan Native	1	2%
Asian	6	10%
Black or African American	29	50%
Native Hawaiian or other Pacific Islander	4	7%
Caucasian	10	17%
Prefer not to answer	8	14%
Hispanic or Latino	21	36%
Current tenure status		
Tenured	2	3%
Tenure track	21	34%
Non-tenure track	13	21%
Not applicable	25	41%
Scholar goal		
Independent investigator	41	67%
Collaborator (not interested in being PI)	3	5%
Clinician	3	5%
Clinician educator/teacher	9	15%
Other	5	8%
Years in research (median, 25th–75th percentile)	8	5–12

^a^ The number of participants across categories may not sum to the total number of participants due to missing data. ^b^ Median and percentiles will be given for variables when indicated.

**Table 2 ijerph-18-06155-t002:** Mentoring structure, characteristics, and effectiveness of mentors and mentoring teams taken from baseline and exit surveys completed by participating scholars.

Mentoring Questions on Survey	Baseline (n = 61)		Exit (n = 26)	
	n	%	n	%
Structure of mentoring relationship				
Primary mentor or predominantly	21	34	11	42
Mentoring team with an individual who as a primary mentor	23	38	13	50
Mentoring team without an individual who serves as a primary mentor	17	28	2	8
Years working with the mentor				
0–3	39	64	16	62
4–7	13	21	5	19
8–10	9	15	5	19
Meetings with the mentor per month				
0–3	46	75	22	85
4–7	10	16	2	8
8–10	5	8	2	8
Overall effectiveness of the mentor ^a^	41	70	19	76
Mentor’s advice about balance of professional and personal life ^a^	27	47	16	64
Mentor is accessible ^b^	49	83	21	84
Mentor demonstrates professional integrity ^b^	55	93	22	85
Mentor provides useful critique of my work ^b^	53	90	22	85
Mentor motivates me to improve my work product ^b^	48	80	21	81
Mentor team contributes more to professional development than the mentor alone ^b^	39	64	18	69
Mentor team provides knowledge and skills that are different from the main mentor ^b^	42	69	20	77
Mentor team effectively communicates about professional development ^b^	39	64	20	77
Mentor helps me develop research skills ^b^	35	60	20	77
Mentor’s advice about career development and balancing professional responsibility ^b^	36	61	22	85
Mentor suggests resources and uses their influence to support my advancement ^b^	46	78	21	84

^a^ (Sc ale 1–5, 5 = Very effective; combined 4 and 5), ^b^ (Scale 1–6, 5 = Strongly agree, combined 4 and 5).

## Data Availability

The data are available upon request from DMR, one of the corresponding authors.

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
