# Peer review of "Challenges and Strategies of Successful Mentoring: The Perspective of LEADS Scholars and Mentors from Minority Serving Institutions"

_ijerph, 2021, doi:10.3390/ijerph18116155_

Round 1

Reviewer 1 Report

A worthwhile study, this project could benefit from an extensive re-write. First, I would encourage you to take a step back and have someone else read through your project for typographical mistakes. Second, consider each statement and the evidence you have to support your ideas.  What is the theoretical premise of your project?  The methods section needs much more information. I have written some comments within your text. Why is a descriptive study the best one for exploring the impact of a mentorship program? 
There might be a better way of presenting your tables.   Review other examples in journals and develop one that will be efficient. 
Discuss your findings concerning other completed studies on mentoring in higher education.  What is your contribution to mentoring scholarship? 

Author Response

Point 1: A worthwhile study, this project could benefit from an extensive re-write. First, I would encourage you to take a step back and have someone else read through your project for typographical mistakes. 

Response 1: We had an editor review the manuscript

Point 2: Second, consider each statement and the evidence you have to support your ideas.  What is the theoretical premise of your project?

Response 2: As this is a descriptive study, we did not have a theoretical framework. We are not positing causal relationships.

Point 3: The methods section needs much more information. 

Response 3: This section has been revised with much more information added.

Point 4: Why is a descriptive study the best one for exploring the impact of a mentorship program? 

Response 4: We now explain this in the beginning of the Methods section

Point 5: There might be a better way of presenting your tables. Review other examples in journals and develop one that will be efficient.   

Response 5: We revised the table to a much-improved format

Point 6: Discuss your findings concerning other completed studies on mentoring in higher education.  What is your contribution to mentoring scholarship? 

Response 6: We included additional references in the Discussion section.

Reviewer 2 Report

The topic addressed is very interesting but the article needs in-depth changes to adjust to scientific standards. In the first section, it would be convenient to introduce recent studies on the subject and consider the pedagogical perspective due to its importance in mentoring. In the Method section, the Participants (Sample), Instruments and Results sections should appear well differentiated and explained in depth. In addition to the description of the data analysis. The results must go well beyond the presentation of the raw data through a table. In Discussion it is convenient to compare the results obtained with previous studies and the Conclusions, due to the importance of the question analyzed, should be developed in depth. In the References section there is an error, for example: some authors appear with their full name and others with initials; the year of the publications appears in different places, with or without parentheses, the cities appear when in APA 7th this information is eliminated.

Author Response

Point 1: In the first section, it would be convenient to introduce recent studies on the subject and consider the pedagogical perspective due to its importance in mentoring. 

Response 1: We included additional articles in the introduction section to address this.

Point 2: In the Method section, the Participants (Sample), Instruments and Results sections should appear well differentiated and explained in depth.

Response 2: The Methods section has been significantly rewritten with more information provided.

Point 3: The results must go well beyond the presentation of the raw data through a table. 

Response 3: As this is a descriptive study, we only present the results of our findings. We did completely revise the tables so that they are more informative. We expanded the Discussion section to have more implications of the findings with supporting studies.

Point 4: In Discussion it is convenient to compare the results obtained with previous studies and the Conclusions, due to the importance of the question analyzed, should be developed in depth.

Response 4: The Discussion section has been expanded with additional references.

Point 5: In the References section there is an error, for example: some authors appear with their full name and others with initials; the year of the publications appears in different places, with or without parentheses, the cities appear when in APA 7th this information is eliminated.   

Response 5: The references have been corrected.

Reviewer 3 Report

The paper is interesting and I think it should be accepted, I think it should be published, although it could be improved if weak aspects are modified in the in the experience´s narration.

Regarding their strengths and weaknesses,  I can indicate the following aspects:

- Areas of the strength of this paper:
.The topic  is interesting (even though mentions of studies on mentoring are lacking in the introduction that could exemplify its claims).

.The paper presents the originality of analyzing a mentoring and professional development program to increase the success of the generation of academics who are underrepresented.
.It is interesting because it addresses the impact that the pandemic had on the program.

. Also is a descriptive study. There are too many works of this style with quantitative studies that do not allow us to really know what the sample says, how it says it,...

 - Areas weakness of this paper:
. The authors do not explain why they chose a descriptive method (even though it is sensed).
. The conclusions are excessively terse. Surely they need to relate them more to the results
. It would be appropriate if the data related to the population characteristics of the scholars studied and the results obtained were represented with graphs for a better understanding or  that  the authors should look for a better way of presenting your data. The tables are unclear.
. The mentor´s profile is also a fundamental piece of information, i think that is necessary in this type of research. It would be necessary for them to delve into the mentors' data.
. The instruments are not  sufficiently explained.
. It is a pity that the paper does not disaggregate the data by sex, the gender perspective is very important in the study of programs, but above all when it comes to underrepresented samples. 

Author Response

Point 1: The authors do not explain why they chose a descriptive method (even though it is sensed) 

Response 1: This is now described in the first paragraph of the Methods section

Point 2: The conclusions are excessively terse. Surely they need to relate them more to the results

Response 2: We expanded the Discussion section and relate the findings to previous studies. We only intended the conclusion to be a summary.

Point 3: It would be appropriate if the data related to the population characteristics of the scholars studied and the results obtained were represented with graphs for a better understanding or that the authors should look for a better way of presenting your data. The tables are unclear. 

Response 3: We completely revised the tables and they are much more informative.

Point 4: The instruments are not sufficiently explained.

Response 4: We revised the Methods section and included much more information, including the instruments.

Point 5: It is a pity that the paper does not disaggregate the data by sex, the gender perspective is very important in the study of programs, but above all when it comes to underrepresented samples.   

Response 5: The majority of the sample are women, so we did not analyse by sex given the few men who participated in LEADS and responded to the surveys.   

Reviewer 4 Report

That was interesting to review the article. The topic is current and very important. Somehow it blends in with the discussion on the broadly understood intergenerational succession.

To increase the article's scientific value I suggest:

  1. clearly define the aim, that will hel to develope the Introduction
  2. to improve the literature analysis 
  3. to clarify the main scope of the article (mentoring at the universities or (academics') career coaching or knowledge/skills intergenerational transfer or sth else)
  4. to develop the Discussion
  5. to describe the research limitations
  6. to analyse barriers others then those related with covid
  7. to develop Conclusions

Author Response

Point 1: clearly define the aim, that will hel to develope the Introduction

ʉ۬

Response 1: The introduction has been revised to articulate the purpose of the paper

Point 2: to improve the literature analysis 

Response 2: This has been revised with more literature added.

Point 3: to clarify the main scope of the article (mentoring at the universities or (academics') career coaching or knowledge/skills intergenerational transfer or sth else) 

Response 3: Please see point 1 above.

Point 4: to develop the Discussion

Response 4: The Discussion section has been revised and we included more studies.

Point 5: to describe the research limitations  

Response 5: We added a section to the Discussion that presents the limitations to the study.

Point 6: to analyse barriers others then those related with covid

Response 6: We focus on COVID because that is the current struggle that the LEADS Scholars are having and we believe most relevant today. There certainly are other barriers or challenges. We included some of those that are more relevant to MSIs such as heavy teaching loads.

Point 7: to develop Conclusions  

Response 7:  We expanded the Discussion section. The Conclusion is only meant to be a summary of the manuscript.

Round 2

Reviewer 1 Report

Unfortunately, Several of the comments I made in the first review were not addressed. 

Also, It will be important to have a native speaker, read through the document and correct the grammar. 

I will reattach the first review for the authors attention. 

Author Response

Response to Reviewer 1 Comments

Point 1: Unfortunately, Several of the comments I made in the first review were not addressed. 

Response 1: Each of these comments are now addressed or reworded in the manuscript.

Point 2: Also, It will be important to have a native speaker, read through the document and correct the grammar. 

Response 2: All but one of the authors are native speakers. 4 native speakers have reviewed the manuscript including an editor.

Point 3: I will reattach the first review for the authors attention. 

Response 3: Please see #1 above.

Reviewer 2 Report

The authors have done a good job and made most of the suggestions but I think the following changes are necessary: - Check the numbering of the authors. - Expand the conclusions. - Review APA standards in References.

Author Response

Response to Reviewer 2 Comments

Point 1: Check the numbering of the authors 

Response 1: We did add an author at the last version, however, the last two authors are from the same institute, therefore the numbers are correct.

Point 2:  Expand the conclusions.

Response 2: Expanded.

Point 3: Review APA standards in References. 

Response 3: Reviewed and corrected.

Reviewer 4 Report

Thank you all Authors for the changes and additions in the manuscript.

Nevertheless I still find the theoretical background of the research as not satisfying enough. There is not sufficiently justified the use of metoring as HR development method/ method in professional career development. It needs to be expanded.

How the results can be used? Are they going to be used for preparing recommendations/ mentoring strategies?

Conclusions needs some development.

Author Response

Response to Reviewer 4 Comments

Point 1: I still find the theoretical background of the research as not satisfying enough. There is not sufficiently justified the use of metoring as HR development method/ method in professional career development. It needs to be expanded.

Response 1: We do not have a theoretical framework as this is a descriptive study. We are not positing any causal relationships.

Point 2: How the results can be used? Are they going to be used for preparing recommendations/ mentoring strategies?

Response 2: We added this to the discussion section.

Point 3: Conclusions needs some development.

Response 3: We completely rewrote the conclusion

This manuscript is a resubmission of an earlier submission. The following is a list of the peer review reports and author responses from that submission.